# Enhanced Succinate Production in *Actinobacillus succinogenes* via Neutral Red Bypass Reduction in a Novel Bioelectrochemical System

**DOI:** 10.3390/biotech14040084

**Published:** 2025-10-29

**Authors:** Julian Tix, Fernando Pedraza, Roland Ulber, Nils Tippkötter

**Affiliations:** 1Bioprocess Engineering and Downstream Processing, University of Applied Science Aachen, 52428 Jülich, Germany; tix@fh-aachen.de (J.T.); fergab.pedraza@gmail.com (F.P.); 2Mechanical and Process Engineering, RPTU Kaiserslautern-Landau, 67663 Kaiserslautern, Germany; roland.ulber@mv.rptu.de

**Keywords:** *A. succinogenes*, succinate, power-to-X, electrofermentation, 2-chamber-system

## Abstract

Carbon capture and power-to-X are becoming increasingly relevant in the context of decarbonization and supply security. *Actinobacillus succinogenes* is capable of transforming CO_2_ into succinate, whereby product formation is significantly limited by the availability of NADH. The aim of this work was to further develop a bioelectrochemical system (BES) in order to provide additional reduction equivalents and thus increase yield and titer. To this end, a new BES configuration was established. A conventional stirred tank reactor (STR) is coupled via a bypass to an H-cell, in which the redox mediator neutral red (NR) is electrochemically reduced and then returned back to the bioreactor. The indirect electron transfer decouples the electrochemical reduction from the biology and results in increased intracellular availability of NADH. The present approach resulted in an increase in yield from 0.64 g·g^−1^ to 0.76 g·g^−1^, corresponding to an increase of 18%. At the same time, a titer of 16.48 ± 0.19 g·L^−1^ was achieved in the BES, compared to 12.05 ± 0.18 g·L^−1^ in the control. The results show that the mediator-assisted, partially decoupled BES architecture significantly improves CO_2_-based succinate production and opens up a scalable path to the use of renewable electricity as a reduction source in power-to-X processes.

## 1. Introduction

Succinate is a dicarboxylic acid produced in the tricarboxylic acid cycle (TCA). It is characterized by two terminal carboxylic acid groups [1,2,3,4]. Due to the complexity of the biological substances used in the production process, biological systems for the production of succinate are much more complicated and sensitive than established chemical processes. This results in a more expensive and therefore less competitive product [5]. However, its biotechnological significance is increasing, particularly in light of the fact that the U.S. Department of Energy has classified succinic acid as one of the twelve most promising bio-based building blocks for the future [6]. Consequently, it is anticipated that growth will increase by 17.8% on an annual basis from 2024 to 2034, attaining a volume of US$1.7 trillion [7]. The biosynthesis of succinic acid can be carried out using natural producers of succinate with high titers, including *Actinobacillus succinogenes* and *Mannheimia succiniciproducens*. These usually have a by nature modified TCA in which succinate is the end product [8,9,10]. In addition, genetically modified organisms such as *Escherichia coli*, *Corynebacterium glutamicum*, and *Saccharomyces cerevisiae* are used for fermentation [11,12,13]. *A. succinogenes* is a facultative anaerobic, capnophilic, Gram-negative microorganism with a pleomorphic appearance [9]. It is characterized by the absence of citrate synthase and isocitrate dehydrogenase, which prevents the incorporation of acetyl-CoA into the oxidative part of the TCA [14]. In the absence of these enzymes, *A. succinogenes* can only utilize the reverse, reductive direction of the TCA. Succinate production is driven by an excess of NAD(P)H within the cell and occurs through the successive reduction in oxaloacetate, enabling the regeneration of NAD^+^, which is required for glycolysis. Particularly the conversion of glyceraldehyde-3-phosphate to D-glycerate-1,3-bisphosphate. Significant succinate production by *A. succinogenes* only occurs in the absence of oxygen, when fumarate is available as the final electron acceptor in *A. succinogenes* metabolism. However, when oxygen is present, it takes over the function of the final electron acceptor and no succinate is produced; the bacterial electron transport chain then functions as usual [15]. When considering succinate production, the balance of reducing agents within the cell is important. Thus, high concentrations of the reducing coenzyme NAD(P)H are required to enable the conversion of PEP to succinate [8]. This is because the conversion of oxaloacetate to malate and the conversion of fumarate to succinate both require a reducing coenzyme [16]. NADH limits succinate formation; accordingly, a shift in the intracellular NAD^+^/NADH ratio in favor of NADH increases succinate production [17]. An overview of the metabolic pathway is shown in Figure 1. The aim of electrofermentation is to feed electrons into the cell by applying an electrical potential, thereby shifting the redox equilibrium. This can be achieved in two different ways. First, electrons are transferred directly from the cathode to membrane-bound proteins (e.g., cytochromes) via direct electron transfer (DET). Alternatively, electrons are transferred via mediator-mediated transfer (MET), whereby shuttle molecules absorb electrons and release them to the cell [18]. Natural electron transporters include primary bacterial metabolites, such as sulfur compounds and H_2_, as well as secondary metabolites, which include flavins [19,20,21]. In addition, exogenous mediators such as NR accelerate electron transfer and enhance coupling to membrane-bound redox proteins [22,23,24]. To achieve the reduction in NADH, voltage must be applied to the system. BES are used for this purpose. They combine microbial metabolism with electrochemistry to convert organic substrates. The reactor design determines the electron flow, product selectivity and efficiency, primarily through the choice between single- and dual-chamber systems [25]. Single-chamber BES (SC-BES) are the simplest and most cost-effective configuration of bioelectrochemical reactors as they do not require membrane-based separation between the anode and cathode [26]. The design described offers a number of advantages. These include easier operation, reduced ohmic resistance, and the possibility of compact reactor designs. These characteristics are particularly relevant in relation to wastewater treatment [27,28]. A significant benefit of these systems is their capacity for straightforward scalability and minimal investment requirements. However, technical challenges arise due to the lack of chamber separation. This phenomenon is known as ‘cross-reactions between anode and cathode processes’, which frequently result in efficiency losses, product instability, and limited product selectivity [29]. The membrane-free design of the apparatus under discussion means that chemical crossover is not prevented [25,30]. Furthermore, oxygen diffusion from the cathode to the anode has been shown to inhibit strictly anaerobic cultures and significantly reduce Coulombic efficiency [31]. In addition, pH gradients occur between the anode and cathode chambers because there is no directed proton conduction, which impairs the activity of sensitive organisms and reduces productivity [32]. The use of *A. succinogenes* as a model organism in SC-BES for the electrofermentative production of succinate is particularly interesting. Recent studies show that coupling an electrical potential modulates the intracellular NAD^+^/NADH balance and can increase succinate yield by up to 15% by Hengsbach et al. in 2024 [1]. In contrast, in dual-chamber BES (DC-BES), the anode and cathode are separated by an ion-permeable membrane, which allows for more precise control of electron flows, higher product purity, and selective recovery of valuable substances, but is associated with higher costs and contamination risks that compromise long-term stability [33]. Membranes are particularly important when high Coulombic efficiencies are required, as they suppress the diffusive crossover of reduced/oxidized species, thereby reducing chemical short circuits and prevent the back-diffusion of products between chambers [25,34]. Among DC-BES configurations, H-cells remain a widely used laboratory platform due to their simplicity and clearly defined transport geometry [19,25].

Based on the reference studies by Park and Zeikus in 1999 [23], we have chosen a combined approach that combines the biological robustness of an STR with the selective, potential-driven electron transfer of a DC-BES. The aim of this study is to specifically exploit the advantages of the DC-BES architecture to maximize intracellular NADH availability and thus succinate formation. In Table 1, we have also provided an overview of various studies in the literature. Unlike previous single- or dual-chamber BES and supported systems, our design introduces a hydraulically decoupled bypass loop between the electrochemical compartment and the STR. This enables selective, potential-controlled electron delivery while maintaining optimal bioprocess conditions, an architecture that has not been reported for *A. succinogenes* to date.

## 2. Materials and Methods

### 2.1. Growth Conditions

The organism *A. succinogenes* DSM 22257 strain 130Z from the Leibniz Institute DSMZ German Collection of Microorganisms and Cell Cultures (Braunschweig, Germany) was used in this study. Two different media were used for the fermentations. First, anaerobic precultures were prepared in serum bottles to provide optimal conditions for the growth of the microorganisms. For this purpose, TSB medium was used, whose pH value was adjusted to 6.8 [38]. The medium was first prepared and then degassed with N_2_ for 20 min to create anaerobic conditions. Fermentation was then carried out for 16 h at 37 °C and 80 rpm. The pre-cultures obtained in this way were then used for further fermentation. For fermentation, a medium described by Wang et al. (2018) and adapted by Tix et al. (2024) was used [35,39]. It contained the following ingredients: 30 g·L^−1^ glucose, 31.5 g·L^−1^ Na_2_HPO_4_·12H_2_O, 10 g·L^−1^ NaHCO_3_, 8.5 g·L^−1^ NaH_2_PO_4_, and 5 g·L^−1^ yeast extract. It should be noted that the glucose fraction was autoclaved separately from the other components of the media. Depending on the respective test conditions, an additional amount of 100 µM NR was added to the medium.

### 2.2. Bioreactor Operation and Experimental Setup

A glass stirred-tank reactor (STR) (Eppendorf Q-series from Eppendorf SE, Hamburg, Germany) with a working volume of 2 L was used to configure the reactor. Furthermore, a bypass was established with an H-cell. The two parts of the cell were separated from each other by a Nafion^TM^ membrane. This resulted in the formation of a cathode chamber, where the reduction process occurred, and an anode chamber. The STR was then connected to the bypass, i.e., the H-chamber, using tubes, as demonstrated in Figure 2. This allowed the medium to circulate from the STR to the bypass and back again. Subsequently, the reduction in the redox mediator occurred within the bypass. Two carbon fabrics electrodes (ACC-5092–15 from Kynol Europa GmbH, Hamburg, Germany) with a surface area of 12 cm^2^ each were placed in the cathode and anode chambers, respectively. This arrangement resulted in an H-cell comprising distinct cathode and anode stack compartments. The carbon fabrics was then connected to the potentiostat (MultiPalmSens4 from PalmSens, Houten, The Netherlands) via platinum wires (Ø = 0.4 mm). An Ag/AgCl reference electrode (SE11NSK7-4 from Meinsberg Sensortechnik, Xylem Analytics Germany Sales GmbH & Co. KG, Weilheim, Germany) was utilized to ensure the maintenance of a stable reference potential. The potential was set to −600 mV. In addition, the STR was kept anaerobic with 80% N_2_ and 20% CO_2_ at a gas supply rate of 0.25 vvm, which also ensured the supply of CO_2_. The H-Cell was continuously supplied with N_2_ gas throughout fermentation in order to maintain anaerobic conditions here as well. *A. succinogenes* was cultivated in the defined main nutrient medium at 37 °C and 150 rpm for 48 h. The pH was maintained at 6.8 by automatic titration with 5 M NaOH. All experiments were conducted as individual cultivation runs with technical triplicates (n = 3). Reported values represent mean concentrations and standard deviation (SD). Data processing and graphical representation were performed using Microsoft Excel (Microsoft Corp., Redmond, WA, USA) and GraphPad Prism 10 (GraphPad Software, San Diego, CA, USA). The standard deviation was calculated to indicate the reproducibility of the measurements. No inferential statistical tests were applied, as the observed trends were consistent across all replicates.

### 2.3. Analytical Procedures

Quantification of soluble substrates and products was performed by high-performance liquid chromatography (HPLC) as described in previously publication [35,40]. Chromatographic separations were performed on a Repromer H^+^ column (300 × 8 mm; Dr. Maisch HPLC GmbH, Ammerbuch, Germany) maintained at 30 °C. Samples (5 µL) were injected and eluted isocratically with 5 mM H_2_SO_4_ as the mobile phase.

## 3. Results and Discussion

The aim of this study was to quantitatively demonstrate the performance gains achievable in CO_2_-based succinate fermentation with the newly designed, bypass-coupled DC-BES. All experiments were performed as technical triplicates. In the DC-BES runs, 100 µM NR was added to the medium and a cathodic potential was applied to activate mediator-mediated electron transfer. Under otherwise identical process conditions, the DC-BES was operated with 100 µM NR at an applied cathode potential of −600 mV (Ag/AgCl). The control ran without electrochemical coupling and without NR to rule out concentration-dependent inhibition by NR in the absence of electrochemical coupling [35,41]. Figure 3 shows that, in the DC-BES, the yield increased from 0.64 to 0.76 g·g^−1^ (Δ = +0.12 g·g^−1^; +18.2%). At the same time, the succinate titer increased from 12.05 ± 0.18 to 16.48 ± 0.19 g·L^−1^ (+37%). The progression of glucose consumption and product formation over time can be seen in Figure 4. After just 24 h, the titer in the DC-BES was 13.35 ± 0.07 g·L^−1^ compared to 7.95 ± 0.20 g·L^−1^ in the control (+68%). After 48 h, the difference was only 37%. This inverse time trend indicates that the stimulatory effect of external electron supply is most pronounced during the initial phase of fermentation. This achieved a volumetric succinate formation rate of 0.54 g·L^−1^·h^−1^ during the first 24 h of operation, demonstrating a high early-phase conversion capacity under electron-supplemented conditions. In the DC-BES, only 3.81 g·L^−1^ glucose remained after 24 h, whereas 9.57 g·L^−1^ was still available in the control, confirming a faster substrate turnover. The enhanced early performance is therefore attributed to the immediate availability of reducing equivalents, which promotes faster metabolic activity and cell growth. As glucose became limiting, the relative difference in titer decreased. This is consistent with (i) an early build-up of a reduced mediator pool due to the immediate availability of external electrons, (ii) increased cell density and mediator turnover during the initial phase, and (iii) a subsequent shift in the intracellular redox balance towards higher NADH availability, which together explain the rapid glucose consumption and the pronounced early succinate formation observed in the DC-BES. The product distributions support this interpretation. In the control, 59% of the product stream was succinate, 17% was formate, and 24% was acetate. In the DC-BES (−600 mV + NR), the ratio shifted to 76% succinate, 6% formate, and 19% acetate. This shift reflects the redirection of intracellular electron fluxes towards the reductive C4 branch of central metabolism (see Figure 1). Under conventional conditions, *A. succinogenes* converts pyruvate into acetyl-CoA and formate via pyruvate formate-lyase (PFL), distributing reducing equivalents between the C3 products acetate, ethanol and formate [42,43]. However, when external electrons are supplied through the bioelectrochemical system, the intracellular NAD^+^/NADH ratio increases. This promotes the reduction in fumarate to succinate via the enzymes fumarate reductase (FRD) and malate dehydrogenase (MDH). Consequently, carbon flux is redirected from the C3 branch towards the C4 branch, leading to increased succinate formation and decreased formate production [42,44]. The lower formate yield indicates that pyruvate is preferentially converted through carboxylation (PEPCK-MDH-FRD pathway) rather than cleavage by PFL. In parallel, the slightly reduced acetate fraction suggests that acetyl-CoA is increasingly oxidized to generate energy instead of being released as acetate, which is consistent with the more reduced intracellular environment created by cathodic electron input [14,45]. Overall, pathway analysis (Figure 1) shows that the DC-BES facilitates an early and sustained shift in redox metabolism towards the reductive succinate-forming C4 pathway.

The marked decline in the formate fraction (−11 percentage points) accompanied by an increase in the succinate fraction (+17 percentage points) suggests a redirection of carbon and electron flow away from formate-forming pathways of mixed acid fermentation toward the reductive TCA branch [8,9,10]. From a mechanistic perspective, this hypothesis appears valid, as the reduction in oxaloacetate to malate requires the formation of NADH. In addition, the subsequent reduction in fumarate to succinate is facilitated by the provision of electrons from a reduced electron donor. An external electron input, mediated by the NR mechanism, could therefore alleviate the bottleneck in NADH availability and increase selectivity in favor of succinate. It is essential that the analysis of the available data be carried out with due care [8,9,35]. The relative increase in titer between 24 h and 48 h was more pronounced than between 0 h and 24 h, suggesting that the system requires a start-up phase before reaching a quasi-steady state. During this time, the mediator cycle stabilizes, mass transfer in the bypass improves, and metabolism adapts to the additional reduction equivalents. The effect is therefore not stronger at the beginning but only becomes more clearly visible after the system has stabilized. The available data suggest that the partially decoupled, mediator-based architecture not only increases yield and final titer but also shifts product selectivity in favor of succinate. This is a key aspect for energy- and carbon-efficient power-to-X bioprocesses. In the early studies conducted by Park and Zeikus on H-cells, it was demonstrated that cathodically reduced NR functions as an electronophore. These findings led to the observation that electron flow in fermentative cultures is facilitated, thereby enabling *A. succinogenes* to reduce fumarate to succinate. So they demonstrated that NR can functionally replace H_2_ as an electron donor [23,46]. In mechanistic terms, it was subsequently elucidated that NR predominantly diminishes the menaquinone reserve of the inner membrane (as opposed to directly impacting NAD^+^) in the context of *E. coli.* This observation provides a rationale for the observed propensity of end products to adopt a more reduced state [47]. Recent electrofermentation studies with *A. succinogenes* have confirmed that an electrically supplied reduction source effectively increases NADH availability, thus increasing succinate yield/titer. Peng et al. report an increase in succinate production of 17% and titers of up to 74.4 g·L^−1^ in fed-batch. The authors also emphasize that control strategy and potential management can significantly improve energy efficiency. It is therefore evident that the process of succinate production is ensured by a high-voltage pulse-based process, while the impact on cell growth and power consumption is minimized. The following essay will provide a comprehensive overview of the relevant literature on the subject [37]. The results described here are thus quantitatively of the same order of magnitude, but are achieved with a partially decoupled bypass architecture, which cleanly separates electrochemistry and biology in terms of process technology. Previously published data suggested that product distribution shifts due to negative potentials (proximity/above-water electrolysis) and succinate yield decreases. This underscores the importance of finely tuned potential windows (e.g., −600 mV). It is essential that the analysis of the available data be carried out with due care [48]. In a single-cell system, it has already been shown that at a potential of −600 mV in comparable SC-BES setups with *A. succinogenes* demonstrated an increase in yield of up to 14% compared to controls [35]. The combination of a DC-BES with a bypass configuration enabled a further increase in yield in this study. In contrast to our strictly bypass-coupled DC-BES design, Pateraki et al. used a modified BES arrangement in which the cathode was in direct contact with the fermentation medium, while the anode was placed centrally in a silicone mesh matrix [36]. Although this geometry creates partial spatial separation, without an ion-selective membrane it is not a conventional DC-BES and is therefore more susceptible to crossover of redox species, and less precise potential control. Under these conditions, Pateraki et al. reported a succinate yield of 0.65 g·g^−1^ without applied potential and, with applied potential, an increase of 6.15% to 0.69 g·g^−1^. In contrast, our DC-BES with a bypass architecture with external mediator reduction specifically addresses these limitations and leads to significantly greater gains while shifting the product distribution in favor of succinate. Compared to conventional electrofermentation, which takes place in the same vessel, the bypass-coupled BES configuration exhibits reproducibly higher yields/titers that are on par with and exceed those of other BES studies. A significant shift in product distribution in favor of succinate was observed.

In addition, process advantages such as selective potential control and lower cross-reactivity were observed at moderate potentials. This suggests a scalable path to power-to-chemical integration for CO_2_-based succinate. To ensure optimized separation, the implementation of a filter between the process steps could be considered. This filter would serve to retain the cells in order to prevent penetration into the H-cell. While the results clearly demonstrate that the DC-BES concept effectively enhances succinate formation through cathodic electron supply, cost effectiveness during scale-up remains a critical aspect. The use of neutral red as a redox mediator substantially contributes to material costs [49,50]. Neutral Red exhibits ecotoxic properties, which complicates large-scale application; however, it serves as an established model mediator for studying bioelectrochemical systems and can be replaced by alternative, more sustainable mediators in future developments [51]. To address this limitation, several strategies can be envisioned: (i) efficient downstream recovery and reuse of neutral red to minimize fresh mediator demand, (ii) substitution with naturally occurring or lower-cost mediators such as safranin or riboflavin, and (iii) further development toward mediator-free electron transfer by coupling the bioreactor to an electrolysis cell generating molecular hydrogen in situ [52,53]. This would enable direct electron uptake via the native hydrogenases of A. succinogenes and eliminate the need for artificial mediators [23]. Although the Nafion^®^ membrane adds to the overall cost, the use of inexpensive carbon fabrics electrodes already improves the economic feasibility of the system. Overall, the results demonstrate a new BES capable of coupling renewable electrical energy with CO_2_ fixation for succinate production. The modular reactor design allows straightforward integration with alternative mediators or mediator-free concepts in future studies, providing a flexible and scalable platform for sustainable power-to-X bioprocesses.

## 4. Conclusions

The present study demonstrates that a bypass-coupled DC-BES architecture with external mediator reduction (−600 mV; 100 µM NR) significantly intensifies CO_2_-based succinate fermentation by *A. succinogenes*. In comparison with the control, yield (0.64 g·g^−1^ → 0.76 g·g^−1^; +18%) and titer (12.05 ± 0.18 g·L^−1^ → 16,48 ± 0.19 g·L^−1^; +37%) increased, whilst product selectivity shifted in favor of succinate (59% → 76%) with a reduced formate content. The time-dependent discrepancy between DC-BES and control demonstrates that decoupled potential control reliably establishes mediator-mediated electron transfer and efficiently addresses intracellular NADH availability as a pivotal bottleneck. In comparison with modulated SC-BES setups with applied potential, the spatial separation of electrochemistry and biology presented here demonstrates greater efficacy under moderate conditions, thus providing a scalable path for integrating renewable electricity into power-to-X bioprocesses. Future research should (i) examine mediator alternatives, (ii) evaluate dynamic potential strategies and control (e.g., pulsed setpoints), and (iii) integrate cell-retaining elements between STR and H-cell to further minimize crossover. The study establishes a precisely controllable, process-robust platform that translates electrical reduction equivalents into higher yields and more selective CO_2_-to-succinate conversion. Future studies should aim to scale up the system, evaluate alternative redox mediators, explore continuous operation, and ultimately enable direct electron transfer via in situ hydrogen generation. Such developments will be essential to fully exploit the technological and industrial potential of this approach.

## Figures and Tables

**Figure 1 biotech-14-00084-f001:**
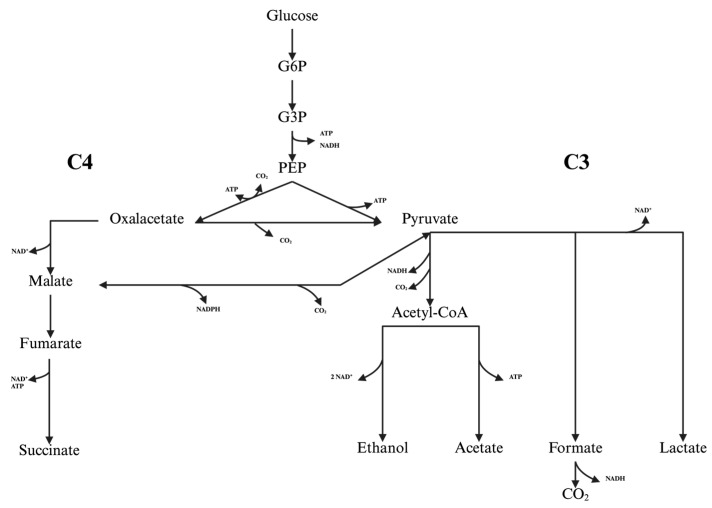
Overview of the central metabolic pathway of *A. succinogenes* starting from glucose as the main substrate. The schematic illustrates the formation of succinate via the reductive C4-branch (PEP → OAA → malate → fumarate → succinate) and the concurrent production of C4 by-products such as ethanol, acetate, formate, and lactate. Modified from Tix et al. (2024) [35].

**Figure 2 biotech-14-00084-f002:**
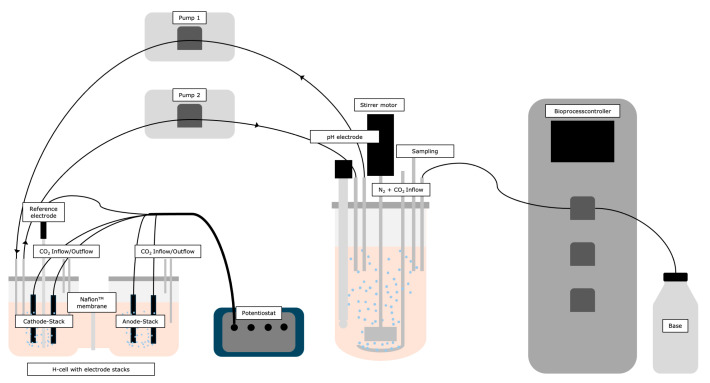
Schematic illustration of Bypass-coupled architecture. A 2 L stirred-tank reactor (STR) is coupled via a bypass loop to a dual-chamber H-cell comprising cathode stack and anode stack compartments. Each compartment contains two carbon fabrics electrodes (ACC-5092–15, Kynol Europa GmbH, Hamburg, Germany; 12 cm^2^ geometric surface area per electrode). NR reduction occurs in the cathode compartment, with culture medium continuously recirculating between the STR and the cathode compartment. An Ag/AgCl reference electrode is positioned in the cathode compartment, and the electrochemical cell is interfaced with a laboratory potentiostat. This configuration indirectly decouples the electrochemical reduction in NR from the biological reactor.

**Figure 3 biotech-14-00084-f003:**
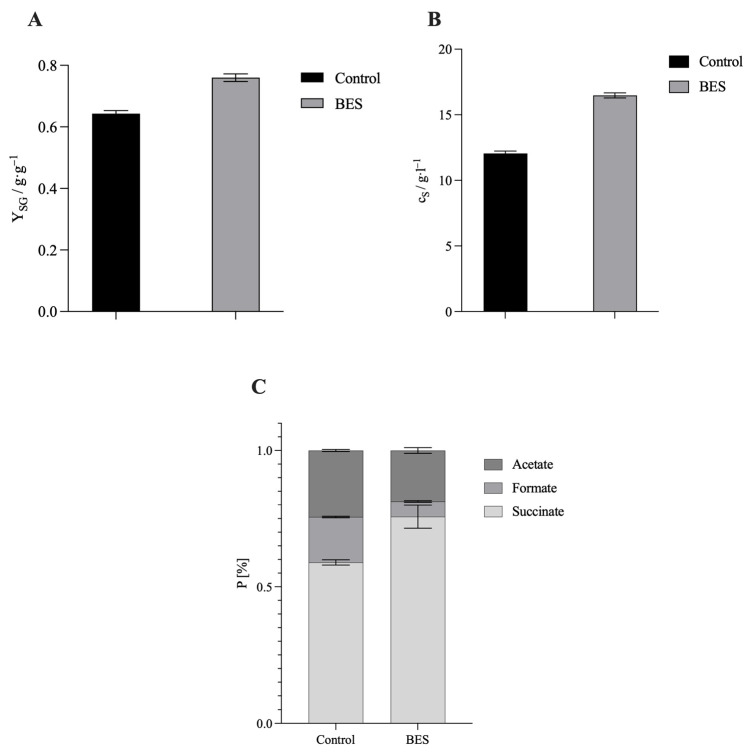
Comparison of reactor runs between control and DC-BES. (**A**) Succinate yield in g·g^−1^ after 48 h plotted. (**B**) Succinate titer after 48 h of fermentation. (**C**) Percentage product distribution after 48 h of fermentation. These are technical triplicates n = 3. Experimental parameters: anaerobic, gassing 0.25 vvm (80% N_2_/20% CO_2_), *A. succinogenes* in the main nutrient medium. V = 2 L, 150 rpm, T = 37 °C, pH = 6.8, pH control with 5 M NaOH, DC-BES with 100 µM NR. Two carbon fabrics electrodes with an area of 12 cm^2^ were installed in each of the cathode and anode chambers. Potential −600 mV in labeled preparations. Bio-Flo 120 bioprocess controller from Eppendorf.

**Figure 4 biotech-14-00084-f004:**
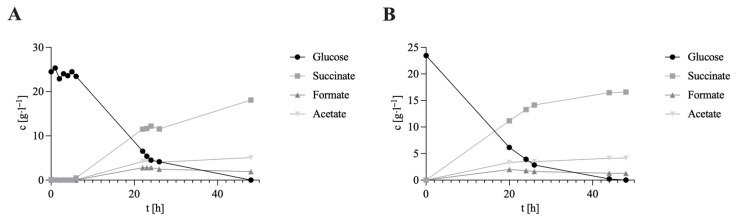
Progression of *A. succinogenes* cultivation under (**A**) control conditions and (**B**) DC-BES operation. Shown are the temporal changes in glucose concentration, and product titers (succinate, acetate, formate, lactate, and ethanol) during 48 h of fermentation. Experimental parameters: anaerobic, gassing 0.25 vvm (80% N_2_/20% CO_2_), *A. succinogenes* in the main nutrient medium. V = 2 L, 150 rpm, T = 37 °C, pH = 6.8, pH control with 5 M NaOH, DC-BES with 100 µM NR. Two carbon fabrics electrodes with an area of 12 cm^2^ were installed in each of the cathode and anode chambers. Potential −600 mV in labeled preparations. Bio-Flo 120 bioprocess controller from Eppendorf.

**Table 1 biotech-14-00084-t001:** Overview of electro-biotechnological studies employing *A. succinogenes* under applied electrical conditions.

Study (Year)	Reactor/Chamber Design	Electric Operation (Potential/Voltage)	Mediator	Notes from Methods
Park & Zeikus (1999) [23]	Two-chamber H-cell (membrane-separated)	Cathodic supply; ~−1.5 V	NR	First demonstration of electrically reduced NR serving as an external electron donor for *A. succinogenes* metabolism.
Pateraki et al. (2023) [36]	BES (no bypass)	Applied potential (exact value n.d.)	n.d.	Investigated transcriptional regulation in *A. succinogenes* under electricity; confirms redox-linked gene expression shifts.
Peng et al. (2024) [37]	Two-chamber MEC	ORP-controlled (−400 mV) and pulsed −1 V (0.5 s on/off)	NR (0.1 mM)	Cathodic enhancement of succinate formation; carbon-felt electrodes, 280 mL per chamber.
Tix et al. (2024) [35]	Single-chamber electro-bioreactor (STR-based)	−600 mV vs. Ag/AgCl	NR (0.1 µM) and more tested	2 L STR; 0.25 vvm (80% N_2_/20% CO_2_); 37 °C; pH 6.8; controlled via BioFlo 120; carbon-fiber electrodes.
Hengsbach et al. (2024) [1]	BES (no bypass)	Controlled potential (~−600 mV range)	NR (0.1 mM)	Investigated potential effects on NAD^+^/NADH ratios; carbon fabric electrodes.

## Data Availability

The original contributions presented in this study are included in the article. Further inquiries can be directed to the corresponding author.

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
