# Peer review of "Enhanced Succinate Production in *Actinobacillus succinogenes* via Neutral Red Bypass Reduction in a Novel Bioelectrochemical System"

_biotech, 2025, doi:10.3390/biotech14040084_

Round 1
Reviewer 1 Report
Comments and Suggestions for Authors
This manuscript presents a study investigating whether the reducing power supplied by a bioelectrochemical system (BES) can be applied to the cultivation of Actinobacillus succinogenes to enhance succinate production. The topic is timely and of potential interest to researchers working in the field of bioelectrochemical systems and microbial bioproduction. However, before the manuscript can be considered for publication, several issues need to be addressed and clarified as outlined below.
-The authors should clearly state how many times experiment was repeated. In addition, details of the statistical methods used to analyze the data should be included to support the validity of the results.
-The discussion on metabolic pathways should be strengthened. Specifically, the manuscript should more explicitly explain the differences in succinate/formate/acetate production ratios between the control experiments and those with DC-BES application. While a brief explanation for the observed increase in succinate and decrease in formate is provided, a more thorough mechanistic interpretation is required. Ideally, a pathway illustration should be included. In particular, the reason for the reduced formate yield needs to be clarified.
-Although some operating conditions (temperature, agitation, duration, etc.) are mentioned in the caption of Fig. 2 and partially in the text, the fermentation conditions in the stirred-tank reactor (STR) must be described precisely in the Materials and Methods section.
-Figure 2 currently shows only calculated values based on cell density and product concentrations. However, presenting the raw time-course data of cell density and product concentrations is essential. The authors should provide separate graphs showing the temporal changes in these parameters.
Comments on the Quality of English LanguageThe manuscript is generally understandable, but careful revision of the English is recommended. Certain words should be replaced with more academically appropriate terms (e.g., “classic” should be revised to “conventional”).
Author Response
We sincerely thank Reviewer for the valuable and constructive comments, which helped us to improve the quality and clarity of our manuscript. In response to the reviewer’s suggestions, we have carefully revised the text, corrected several numerical values, improved the English language throughout the manuscript, and added new data and figures as described below.
Comment:
-The authors should clearly state how many times experiment was repeated. In addition, details of the statistical methods used to analyze the data should be included to support the validity of the results.
Response:
We have now added this information to the Materials and Methods section. The former Section 2.1 was revised and renamed to ‘2.1 Bioreactor operation and experimental setup’.
Comment:
-The discussion on metabolic pathways should be strengthened. Specifically, the manuscript should more explicitly explain the differences in succinate/formate/acetate production ratios between the control experiments and those with DC-BES application. While a brief explanation for the observed increase in succinate and decrease in formate is provided, a more thorough mechanistic interpretation is required. Ideally, a pathway illustration should be included. In particular, the reason for the reduced formate yield needs to be clarified.
Response:
We appreciate this valuable comment. To address it, we added a metabolic pathway overview as Figure 1 in the Introduction, illustrating the central C₄- and C₃-pathways of Actinobacillus succinogenes. Furthermore, the Discussion section has been substantially expanded to provide a more detailed mechanistic interpretation of the observed product ratios, supported by relevant literature. The revised text now explains that the increase in succinate and the decrease in formate are caused by a redirection of electron fluxes toward the reductive C₄-branch under electron-rich conditions.
Comment:
-Although some operating conditions (temperature, agitation, duration, etc.) are mentioned in the caption of Fig. 2 and partially in the text, the fermentation conditions in the stirred-tank reactor (STR) must be described precisely in the Materials and Methods section.
Response:
Thank you for this important remark. The detailed fermentation parameters (working volume, gas composition, agitation speed, temperature, pH control, electrode potential, and mediator concentration) have been added to the newly revised Section 2.1 Bioreactor operation and experimental setup. This addition ensures that the operating conditions are now comprehensively described and fully reproducible.
Comment:
-Figure 2 currently shows only calculated values based on cell density and product concentrations. However, presenting the raw time-course data of cell density and product concentrations is essential. The authors should provide separate graphs showing the temporal changes in these parameters.
Response:
In the revised manuscript, we have included an additional figure (Figure 4) showing the complete time-course data for both control (Figure 4A) and DC-BES (Figure 4B) experiments. The figure presents the temporal evolution of cell density, glucose consumption, and product formation (succinate, acetate, formate, lactate, and ethanol) during 48 h of cultivation. This addition visualizes the dynamic differences between both conditions and complements the previously shown calculated yields.
The manuscript is generally understandable, but careful revision of the English is recommended. Certain words should be replaced with more academically appropriate terms (e.g., “classic” should be revised to “conventional”).
Great. Thank you very much. We have adjusted that.
Reviewer 2 Report
Comments and Suggestions for Authors
This paper discusses CO₂-based succinate production using Actinobacillus succinogenes, specifically suggesting a method to enhance productivity by increasing NADH availability through a novel bioelectrochemical system (BES) with a bypass structure. However, the paper requires some revisions before publication.
1. What is the succinate conversion capacity of the lab-scale electrochemical system shown in Figure 1? Is sustainable use possible? If the available capacity of the lab-scale treatment device is limited, improvements are needed to improve process efficiency. This issue ultimately relates to the stability and reproducibility of the treatment system.
2. Overall, the lack of figures and tables does not adequately convey the authors' intended results, and the novelty of this study compared to previous studies is lacking. More research findings are needed, and a redesign of the study design may be necessary.
3. The products consist of succinate, formate, and acetate. The authors increased succinate yield to 76% using an electrochemical conversion method. However, the basic metabolic environment is limited by the three-product composition. Therefore, significant improvements in succinate yield are unlikely. The significance of the improvements achieved in this paper needs to be demonstrated.
4. A table comparing yields (conversion rates) with other existing succinate production methods, production conditions, energy requirements, etc., should be included to effectively demonstrate the uniqueness of this study.
Author Response
We sincerely thank the Reviewer for the valuable and constructive comments, which helped us to improve the quality and clarity of the manuscript. In addition to the specific revisions detailed below, we slightly adjusted some numerical values according to most recent experimental repetitions and re-calculations of experimental data to ensure accuracy and consistency throughout the manuscript.
Comment 1:
- What is the succinate conversion capacity of the lab-scale electrochemical system shown in Figure 1? Is sustainable use possible? If the available capacity of the lab-scale treatment device is limited, improvements are needed to improve process efficiency. This issue ultimately relates to the stability and reproducibility of the treatment system.
Response
Thank you for this important comment. In the revised version, we have provided additional details regarding the performance of the DC-BES system. Specifically, the volumetric succinate formation rate was determined to be 0.54 g·L⁻¹·h⁻¹ during the first 24 h, demonstrating the high conversion efficiency in the early fermentation phase under electron-supplemented conditions. This new information has been added to the Results and Discussion section.
Furthermore, we expanded the discussion on the scale-up potential and sustainability of the system. While neutral red was used as an effective model mediator for laboratory-scale proof-of-concept experiments, its long-term cost and ecotoxicity limit industrial applicability. Possible solutions include mediator recovery, substitution with naturally occurring compounds such as safranin O or riboflavin, and the development of mediator-free configurations based on in situ hydrogen generation. These aspects are now discussed in the revised manuscript to address the reviewer’s concern.
Comment 2:
- Overall, the lack of figures and tables does not adequately convey the authors' intended results, and the novelty of this study compared to previous studies is lacking. More research findings are needed, and a redesign of the study design may be necessary.
Response
To better present our findings and emphasize the novelty of our work, we have added Table 1, which summarizes previous electro-biotechnological studies on Actinobacillus succinogenes under applied electrical conditions. This allows direct comparison and highlights the unique design of our bypass-coupled DC-BES system, which separates the electrochemical and biological compartments while maintaining controlled mediator-assisted electron transfer.
Additionally, Figure 4 was added to show the temporal profiles of succinate concentration, glucose consumption, and cell density during both control and DC-BES runs. These changes improve data visualization and clearly demonstrate the enhanced early performance and substrate utilization of the DC-BES setup.
Comment 3:
- The products consist of succinate, formate, and acetate. The authors increased succinate yield to 76% using an electrochemical conversion method. However, the basic metabolic environment is limited by the three-product composition. Therefore, significant improvements in succinate yield are unlikely. The significance of the improvements achieved in this paper needs to be demonstrated.
Response
We thank for this helpful observation. The discussion on the metabolic background and the mechanistic interpretation of product ratios has been expanded. A new metabolic pathway overview (Figure 1) was added to the Introduction, illustrating the central carbon flux from glucose to succinate and by-products. In the revised Discussion, we now explicitly address the changes in the succinate/formate/acetate distribution between the control and the DC-BES (76 % / 6 % / 19 %), and explain the observed reduction in formate yield through the altered NADH/NAD⁺ balance and mediator-driven electron uptake. This provides a clearer mechanistic understanding of the results.
Comment 4:
- A table comparing yields (conversion rates) with other existing succinate production methods, production conditions, energy requirements, etc., should be included to effectively demonstrate the uniqueness of this study.
Response
We thank the reviewer for this suggestion. In the revised manuscript, we have included Table 1, which summarizes previously published electro-biotechnological studies involving Actinobacillus succinogenes. The table compares the respective reactor configurations, applied electrical conditions, and mediators used, thereby providing a clear overview of the methodological landscape and highlighting the distinct features of our DC-BES setup.
Quantitative succinate yield data were not consistently available across the reviewed studies and could therefore not be uniformly incorporated into the table. Instead, yield and productivity aspects are discussed in more depth within the Results and Discussion section, where we directly compare representative studies with our findings to illustrate the performance and uniqueness of the presented system.
Reviewer 3 Report
Comments and Suggestions for Authors
The article is well written.
Points to be improved:
- figure quality (fig 1). More detail is needed.
- cost effectiveness of this system during scale up may be challenging. Solution can be stated in the discussion
- environmental impact of NR in fermentation process should be stated
Author Response
Comment 1:
figure quality (fig 1). More detail is needed.
Response
We thank for this helpful comment. In the revised manuscript, the former Figure 1 has been updated and is now presented as Figure 2 due to the addition of a new preceding figure. The schematic has been improved to provide a clearer and more complete representation of the experimental setup. Specifically, we added the second peristaltic pump used for recirculating the medium from the external loop back into the stirred-tank reactor (STR). In addition, the potentiostat and the electrical connections to the electrode stacks, including the reference electrode, have been incorporated and labeled. Two further designations were included for the potentiostat and the Nafion® membrane to improve technical clarity. We hope that these additions sufficiently address the reviewer’s comment for a more detailed and informative figure. Should the reviewer have additional suggestions regarding specific elements to be included, we will be pleased to implement them.
Comment 2:
cost effectiveness of this system during scale up may be challenging. Solution can be stated in the discussion
Response
We appreciate this valuable remark. Since the present study represents a laboratory-scale proof of concept, neutral red (NR) was employed as a model mediator for controlled benchmarking of the DC-BES performance. In the revised manuscript, the Discussion section now includes a comprehensive paragraph addressing cost and scalability aspects. Specifically, we highlight possible strategies for NR recovery and reuse, discuss potential alternative mediators such as safranin and riboflavin, and evaluate the contribution of inexpensive carbon-fiber electrodes to overall cost efficiency during scale-up. Furthermore, we have added a short paragraph on the environmental impact of NR and discussed future options for replacing it with more sustainable mediators. These additions clarify that, although NR is cost-intensive and ecotoxic, it serves as a useful model mediator whose function can be substituted in future developments.
Comment 3:
environmental impact of NR in fermentation process should be stated
Response
Thank you for pointing out this important aspect. We have added a dedicated statement in the Discussion addressing the ecotoxic properties of neutral red. However, it represents a valuable model mediator for BES research and can be replaced by other, more environmentally benign compounds in future system designs.
Round 2
Reviewer 1 Report
Comments and Suggestions for Authors
A manuscripts is revised correspond to reviewer's suggestions.
Reviewer 2 Report
Comments and Suggestions for Authors
This paper presents a method (DC-BES) to address the issue of NADH deficiency limiting succinate production in A. succinogenes-based CO₂ fermentation. While the initial paper had some shortcomings in data and logic, the authors generally incorporated reviewers' primary comments well in revising the manuscript.
- Added production rate (0.54 g/L/h) and metabolic pathway diagram
- Updated Table 1 (comparison with previous studies), Figures 1 and 4
- Enhanced explanation of why the improvement in this study (76% succinate) is significant and its mechanism
- Enhanced discussion of industrialization prospects
- Updated other data
Therefore, this paper is deemed publishable.